# Evaluation of Probiotic *Bacillus velezensis* for the Control of Pathogens That Cause Post-Weaning Diarrhea in Piglets—Results from In Vitro Testing and an In Vivo Model Using *Caenorhabditis elegans*

**DOI:** 10.3390/microorganisms13061247

**Published:** 2025-05-28

**Authors:** Pia Bilde Rasmussen, Josh Walker, Stacey Robida Stubbs, Andreea Cornelia Udrea, Chong Shen

**Affiliations:** 1Gut Immunology Laboratory, R&D, Health & Biosciences, IFF, Edwin Rahrs Vej 38, 8220 Brabrand, Denmark; pia.bilde.rasmussen@iff.com (P.B.R.); andreea.cornelia.udrea@iff.com (A.C.U.); 2Direct-Fed Microbials Laboratory, R&D, Health & Biosciences, IFF, Nutrition Biosciences USA 1, LLC, 200 Powder Mill Road, Experimental Station—E361, Wilmington, DE 19803, USA; joshua.walker@iff.com; 3C. elegans Laboratory, R&D, Health & Biosciences, IFF, Nutrition Biosciences USA 1, LLC, 200 Powder Mill Road, Experimental Station—E361, Wilmington, DE 19803, USA; stacey.i.robida-stubbs@iff.com

**Keywords:** *Bacillus velezensis*, pathogenic bacteria, gut health, porcine intestinal epithelial cells, probiotic, swine

## Abstract

We investigated the effect of probiotic *Bacillus velezensis* strains (LSSA01, 15AP4 and 2084) on pathogens causing post-weaning diarrhea in piglets (Enterotoxigenic *Escherichia coli*, *Clostridium perfringens*, *Salmonella* spp.). We studied the effect of *B. velezensis* and its cell-free supernatant on (1) pathogen growth; (2) IPEC-J2 cell cytokine and tight junction protein expression; (3) IPEC-J2 cell ‘wound’ recovery; (4) adhesion to IPEC-J2 cells and pathogen exclusion; and (5) *Caenorhabditis elegans* survival following pathogen exposure. Cell-free supernatant (CFS) from all strains inhibited the growth of ETEC F4 and F18 (by 36.9–53.2%; *p* < 0.05). One or more strains inhibited *C. perfringens* and *Salmonella* spp. (*p* < 0.05). Strain 2084 CFS increased IL-8 expression (+12.0% vs. control; *p* < 0.05; 6 h incubation), whereas LSSA01 CFS increased the expression of tight junction proteins (*p* < 0.05 vs. control; 6 h incubation) and accelerated 96 h ‘wound’ healing. Colony-forming units (CFUs) of all strains displayed a higher binding affinity to IPEC-J2 cells than 12 ETEC isolates, reduced adhesion of ETEC F4 and F18 and extended *C. elegans* survival over 30 d. The results indicate that probiotic *B. velezensis* strains have potential for use in the control of PWD pathogens.

## 1. Introduction

Post-weaning diarrhea (PWD) is one of the most economically important diseases affecting swine production worldwide [1]. Changes in diet composition and nutrient levels that occur around the time of weaning alter the intestinal microbiome and can lead to insufficient development of gut mucosal immunity, especially if piglets are weaned early [2]. This results in impaired barrier function, which can predispose piglets to diarrhea and enteric infections [3].

Several microbial pathogens cause PWD, including *Campylobacter* spp., *Clostridium perfringens*, *Escherichia coli*, *Salmonella* spp., group A rotavirus (RV-A) and coronaviruses (transmissible gastroenteritis virus—TGEV; porcine epidemic diarrhea virus—PEDV), as well as nematode and protozoan parasites [4]. Enterotoxigenic *Escherichia coli* (ETEC) has been identified as one of the most prevalent causes of PWD. The ETEC bacterium adheres and subsequently colonizes the small intestine using the F4 (K88) and F18 specific fimbriae and their receptors. Heat-labile toxin (LT) and heat-stable toxins A (STa) and B (STb) are the main enterotoxins detected in swine ETEC [5].

Alternatives to antibiotics or zinc oxide for the control of PWD are needed due to recent regulatory changes in major regions (such as the EU) that have banned their use for growth promoter purposes. Candidate alternatives have been investigated intensively, including essential oils, enzymes, plant extracts, organic acids and probiotics [6]. Probiotics (also termed direct-fed microbials, DFM) are defined as “live microorganisms which when present in adequate amounts confer health benefit on the host” [7]. In farmed pigs, the frequently used probiotics are *Lactobacillus*, *Bifidobacterium*, *Enterococcus*, *Streptococcus* and yeasts from the genus *Saccharomyces* [8]. The addition of *Lactobacillus*, *Bifidobacterium* or *Bacillus subtilis* to piglet diets has been shown to reduce piglet mortality, the prevalence of *E. coli* in feces and the occurrence of diarrhea and to increase the production of short-chain fatty acids (SCFAs) in the cecum [8]. Similar findings have been reported in ETEC challenge models [9]; *Pediococcus acidilactici*, *Saccharomyces* and *Lactobacillus plantarum* appear to limit the attachment of ETEC and reduce the incidence of diarrhea [10,11]. *Bacillus*-based probiotics have been reported to promote a healthy gastrointestinal microbiota and enhance growth in swine [12,13]. The modes of action of probiotics in the piglet gut are still being elucidated. However, current understanding is that they include the following: improving gut barrier integrity, maintaining the natural intestinal microflora by competitive exclusion and antagonism of pathogens, production of bacteriocins with bacteriocidal effects, altering nutrient metabolism by increasing digestive enzyme activity and decreasing bacterial enzyme activity and ammonia production, improving feed intake and digestion, neutralizing enterotoxins and stimulating the immune system [14,15].

*Caenorhabditis elegans* is a small, non-parasitic, free-living, soil nematode that feeds on bacteria. *C. elegans* has many desirable traits that make it an attractive animal model for studying host–pathogen interactions, including its small size, ease of propagation and maintenance, short lifecycle and lifespan, compact and fully annotated genome and wide availability of genetic tools [16]. *C. elegans* has been used previously as an animal model for studies of aging [17] and innate immunity [18,19]. It is increasingly being used to study the effect of probiotics on pathogen–host interactions [20]. In particular, under pathogen challenge, probiotic *Lactobacillus acidophilus* can enhance host resilience as evidenced by an extended *C. elegans* lifespan [21].

In this study, we used a combination of in vitro assays and an in vivo (*C. elegans*) model to study the probiotic modes of action of three strains of *Bacillus velezensis* against common pathogens that cause PWD. The three test strains LSSA01, 15AP4 and 2084, together with another control strain 27, are all from IFF proprietary probiotic strains and their identities were confirmed and described previously [22,23,24,25]. The aim was to evaluate the potential of the probiotic strains as an alternative solution for the control of pathogens that cause PWD. The three *B. velezensis* strains have previously been shown to be efficacious in poultry for stabilizing the gut microflora, improving gut barrier integrity and improving growth performance [22,23,24] and in vitro for reducing the pathogenesis of poultry pathogens [25]. However, their effects on swine host–pathogen interactions have not previously been evaluated. The hypothesis for the in vitro assays was that the probiotic bacteria would confer beneficial effects by pathogen inhibition, exclusion, reduction in adhesion, promotion of wound healing or enhanced expression of gut immunity and integrity biomarkers. The hypothesis for the in vivo *C. elegans* model testing was that the probiotic strains would extend *C. elegans* survival and thereby reduce mortality.

## 2. Materials and Methods

### 2.1. Reagents and Materials

All cell culture media, equipment and reagents were purchased from Thermo Fisher Scientific (Rosklide, Denmark) unless otherwise stated.

### 2.2. Cell Line, Bacterial Strains, Culture Conditions and Preparation of Cell-Free Supernatant

The porcine IPEC-J2 cell line was purchased from DSMZ (Braunschweig, Germany), cultured in Dulbecco’s Modified Eagle Medium (high-glucose DMEM, GlutaMAX™ Supplement), supplemented with 20% fetal bovine serum (FBS) and 1% penicillin/streptomycin (P/S; penicillin 100 units (U)/mL and streptomycin 100 µg/mL), and cultivated at 37 °C anaerobically in an atmosphere of 5% CO_2_. Four *B. velezensis* strains were used in the experiments: Strains LSSA01, 15AP4 and 2084 are proprietary strains that together form part of Enviva^®^ Pro (Danisco Animal Nutrition & Health, IFF, Oegstgeest, The Netherlands), which is used in poultry, and were supplied in-house. LSSA01 and 15AP4 were isolated from turkey litter by Agtech Products (Manhattan, KS, USA) in 2002 and 2000, respectively; 2084 is a non-proprietary commercial strain from Novonesis (Bagsværd, Denmark) and the Microbial Discovery Group (Oak Creek, WI, USA); and Strain 27 is a non-proprietary commercial strain originally purchased from Semco Bioscience (Milwaukee, WI, USA) in 2004, which was used to compare against the inhibitory effect of the probiotic strains. The ETEC, *C. Perfringens* and *Salmonella* isolates were obtained from the Danisco Global Culture Collection (DGCC; Niebüll, Germany). All isolates originated from pig farms were genome sequenced and are identified in Appendix A. Inoculum (agar) plates of the ETEC and *Salmonella* spp. isolates were grown on Tryptic Soy Agar (TSA) at 37 °C under aerobic conditions, while the *C. perfringens strain* was grown on Brain–Heart Infusion (BHI) agar at 37 °C under anaerobic conditions (using Oxoid AnaeroGen sachets). The *B. velezensis* strains were grown aerobically in Tryptic Soy Broth (TSB) at 37 °C until an optical density (OD) of 0.25–0.5 (measured at 600 nm) was reached. Absorbance was measured using a Synergy MX Microplate Reader (BioTek Instruments, Inc., Highland Park, VT, USA). The *Bacillus* bacterial cultures were then centrifuged, and they were sterilized by filtering through a 0.2 µM vacuum filter (Thermo Scientific Nalgene Filter Unit, 500 mL, 0.2 μm aPES Membrane, 75 mm diameter, #566-0020) to obtain cell-free supernatant (CFS). The CFS was stored at −20 °C until further use.

### 2.3. Pathogen Inhibition Assay

In the pathogen inhibition assay, ETEC and *Salmonella* spp. were grown in Tryptic Soy Broth (TSB) at 37 °C under aerobic conditions, and *C. perfringens* was grown in BHI + 0.5 g/L Yeast Extract + 0.05 g/L L-Cysteine (BHI + YE + LC − Teknova #B9970) at 37 °C under anaerobic conditions. After 24 h, a 200 μL sample of the resulting broth was taken and used to inoculate a bacterial colony into a 96-well microplate (Corning #3474). The microplates were then incubated in a shaking incubator for 4 h at 37 °C and 200 rpm. Microplates containing ETEC or *Salmonella* spp. were incubated aerobically, and microplates containing *C. perfringens* were incubated anaerobically, in this step and in subsequent steps. The resulting cultures were used to inoculate a separate microplate containing culture media and CFS from *B. velezensis* strain LSSA01, 15 AP4, 2084 or 27, or culture media alone (as a control). For incubations of *B. velezensis* CFS with ETEC and *Salmonella* spp., 20% *v*/*v* of CFS and 1% *v*/*v* of inoculum culture were used. For incubations with *C. perfringens*, 5% *v*/*v* of CFS and 4% *v*/*v* of inoculum culture were used. The total volume per well was 200 μL. After inoculation, the microplates were incubated at 37 °C for 16 h. The absorbance was then read at an OD of 600 nm, and the percentage inhibition of pathogen growth was calculated according to the following equation:% inhibition = [(1 − (ODa − OD0)/(ODb − OD0))] × 100(1)
where
ODa = OD600 of pathogen plus CFS, after 16 h of incubation;OD0 = OD600 of media alone, after 16 h of incubation (no inoculum and no CFS);ODb = OD600 of the pathogen and culture media alone, after 16 h of incubation (no CFS).

Each experiment was performed three times with three technical replicates. The following pathogen isolates were tested: four F4^+^ and five F18^+^ ETEC isolates, seven Type A, five Type C and two non-Type A or Type C *C. perfringens* isolates, six *S. typhimurium* isolates, two *S. livingstone* isolates and five other *Salmonella* spp. isolates (as indicated in Appendix A).

### 2.4. IPEC-J2 mRNA Expression Without or with Probiotic CFS Pretreatment

Porcine IPEC-J2 cells were seeded in 96-well cell culture plates at a density of 2 × 10^4^ cells/mL in DMEM, with a total volume of 200 µL per well. Cells were differentiated for 2 days at 37 °C until confluency. On day 2, CFS from *B. velezensis* (LSSA01, 15AP4 or 2084) was added to reach a final concentration equivalent to that produced by 1 × 10^7^ CFU/mL per well and the cells incubated for 6 h at 37 °C in an atmosphere of 5% CO_2_. Cells (IPEC-J2) in culture medium with 10% TSB media served as the control. After incubation, the cells were washed with phosphate-buffered saline (PBS) and lysed in 200 µL RLT buffer (Qiagen, Hilden, Germany) for 20 min, before adding proteinase K (Merck) to reach a final concentration of 100 µg/mL. The lysate was then harvested, and the expression of cytokines and tight junction proteins analyzed by reverse-transcription quantitative polymerase chain reaction (RT-qPCR). The extraction of RNA and the RT-qPCR were performed by BioXpedia (Åarhus, Denmark) using the primers listed in Appendix A and according to the method previously described by Shen et al. [25]. The percentage change in the expression of cytokines IL-6, IL-8 and IL-10 and of tight junction proteins zonula occludens-1 (ZO-1), ZO-2, ZO-3, claudin-1, claudin-3, claudin-4 and occludin (OCLN) was determined as described by Shen et al. [25].

Prior to analysis, data were normalized to two sets of housekeeping genes using the following equation:Value = 2 − (Ct sample − Ct housekeeping) × 10^3^(2)

Experiments were performed twice with four replicates per experiment (eight replicates in total).

### 2.5. Wound-Healing Assay

Porcine IPEC-J2 cells were seeded in culture-insert 2 wells (Culture-insert 2 well in µ-Dish 35 mm; IBIDI) and cultured at 37 °C for 72 h until 100% confluency. A ‘wound’ was then made by removing the insert in the middle of the two wells, which covers an area of 64 mm^2^. Cell-free supernatant from *B. velezensis* LSSA01 was then added to each 2 well to reach a final concentration equivalent to that produced by 1 × 10^6^ CFU/mL. A commercial blend of thymol and cinnamaldehyde (Enviva^®^ EO; Danisco Animal Nutrition & Health, IFF, Oegstgeest, The Netherlands), added to experimental wells at 100 ng/mL in place of CFS, was used as a positive control, as previously described by Shen C et al. [26]. Cells in culture medium with 10% TSB media served as the negative control. The ‘healing’ process in each 2 well was observed after each 24 h period for a total of 96 h, using a normal inverted microscope and the recovery area (as a percentage of the total wound area) calculated according to the following equation:Recovery area (%) = recovered area in mm^2^/64 mm^2^ × 100%(3)

The data were analyzed to calculate the percentage difference in recovery area compared to the control, according to the following equation:

Recovery area difference vs. control (%) = Recovery % in the sample with CFS − Recovery % in the sample without CFS

Experiments were performed twice with three replicates per experiment.

### 2.6. Adhesion, Exclusion and Competitive Exclusion Assays

**Bacterial adhesion:** The adhesion assay was performed as previously described by Shen et al. [26]. In brief, IPEC-J2 cells were seeded in 96-well plates at a density of 2 × 10^4^ cells/well in a total volume of 0.2 mL and grown for two days until 100% confluency. *B. velezensis* strains (LSSA01, 15AP4 and 2084) and ETEC bacterial isolates (EC-88, EC-90 and EC-23) were grown separately in TSB at 37 °C under aerobic conditions for 48 h, the OD measured at 600 nm and the concentration adjusted to an OD of 1.00. A 30 μL sample of ETEC or *B. velezensis* cells was then added directly onto the IPEC-J2 cells in each well in 96-well plates and the cells co-cultured for 30 min at 37 °C under aerobic conditions (CFU_loaded_). The cell monolayer was then washed five times with PBS and the cells lysed with cold 0.1% Triton X-100 solution. The lysates were then serially diluted (10-fold) in PBS and plated on TSA for the enumeration of adherent bacteria (CFU_adhered_). Plates were cultured at 37 °C for 24 h before colony counting. For the identification of bacteria loaded (CFU_loaded_), the bacterial suspension prior to its addition to the IPEC-J2 cells was also serially diluted in parallel and plated onto TSA plates. The percentage adhesion of ETEC cells or *B. velezensis* to the IPEC-J2 cells was calculated according to the following formula:Cell adhesion (%) = [(CFU_adhered_)/(CFU_loaded_)] × 100(4)

The data are means from two experiments with three replicates per experiment.

**Pathogen exclusion and competitive exclusion:** Both the exclusion assay (capacity of *B. velezensis* CFU to exclude ETEC from adhering to IPEC-J2 cells when applied to cells before ETEC application) and competitive exclusion assay (capacity of *B. velezensis* to outcompete ETEC from adhering to IPEC-J2 cells when applied simultaneously) were performed in a similar manner to that described by Kadekar et al. [27], but adapted for study in porcine cells. In brief, in the exclusion assay, IPEC-J2 cells were seeded on plates at a density of 2 × 10^4^ cells/well and grown to 100% confluency. Fresh cultures of *B. velezensis* (LSSA01, 15AP4 and 2084) were prepared, their OD measured and adjusted to an OD of 1.00 and the bacteria loaded directly onto IPEC-J2 cells as described for the adhesion assay. The test ETEC isolates (12 F4^−^F18^+^ isolates, as listed in Appendix A) were grown in culture media and the OD of each culture adjusted to an OD of 1.00 as for the *B. velezensis* cultures. A 300 µL sample of each resulting ETEC culture was then centrifuged and the pellet resuspended in 30 μL of culture medium; then, the resulting suspension was added directly to the *B. velezensis* IPEC-J2 cell co-culture without a washing step. For the competitive exclusion assay, *B. velezensis* and ETEC strain cultures were prepared in the same way and loaded simultaneously onto IPEC-J2 cells. In both experiments, the cells were then incubated at 37 °C for 30 min under aerobic conditions. After incubation, the IPEC-J2 cells were washed (five times with PBS) and lysed with 0.1% Triton-X solution, as before. The lysates were serially diluted (10-fold) and loaded onto either TSA plates (for the enumeration of ETEC and *B. velezensis*, CFU_ETEC+_ CFU_bacillus_) or MacConkey plates (for the enumeration of ETEC, CFU_ETEC_). Plates were cultured as for the adhesion assay, the CFUs counted (CFU_adhered_) and the percentage of adhered cells calculated, as in the cell adhesion assay.

The reduction in the percentage adhesion (here termed ‘exclusion’) in the *B. velezensis*-treated groups compared to the response in the medium control (IPEC-J2 cells with ETEC but without *B. velezensis*) was calculated according to the following equation:Exclusion (%) = [1 − (%ETEC adhered_with *B. velezensis*_/%ETEC adhered_without *B. velezensis*_] × 100(5)
where the percentage ETEC adhered was calculated according to the following equation:% ETEC adhered= (CFU_ETEC_/(CFU_ETEC+_ CFU_bacillus_)) × 100(6)

Experiments were performed twice with seven replicates in total.

### 2.7. Nematode-Killing Assay

*C. elegans* strain NL2099, genotype rrf-3(pk1426), was obtained from the Caenorhabditis Genetics Center (Minneapolis, MN, USA) and cultured at 15 °C on nematode growth medium (NGM) plates seeded with a lawn of *E. coli* strain OP50 cultured in Luria Bertani (LB) broth. Animals were synchronized by bleaching and allowed to develop to the late L4 stage on UV-killed OP50 at 25 °C. The *E. coli* OP50 were UV-killed by treating the OP50-seeded plates in a UV crosslinking box (Agiltron Inc., Woburn, MA, USA) at 254 nm for 20 min. A loop of bacteria from the UV-treated plates was streaked onto an LB plate and grown at 37 °C to confirm that they had been killed. Plates exhibiting no bacterial growth after 24 h were used in subsequent experiments.

For the nematode-killing assay, NGM plates were seeded with either *E. coli* OP50, *E. coli* 33 or *B. velezensis* LSSA01, 15AP4 or 2084. These cultures were grown overnight in LB at 37 °C and then a 500 μL sample was taken and spotted onto NGM plates and the plates allowed to dry. Approximately 120 late-stage L4 worms were then placed onto the dried plates and the plates incubated for 48 h at 25 °C. Worms were then transferred to killing assay plates (approximately 20–30 worms per plate). Killing assay plates were prepared as follows: ETEC (strain F4 and F8) and *Salmonella* (*S. typhimurium* and *S. livingstone*) isolates were grown aerobically in TSB overnight at 37 °C, while *C. perfringens* (type A and type C) isolates were grown anaerobically in BHI overnight at 37 °C. Cultures were then concentrated 1:2 in their respective growth media and 200 μL of the resulting culture was seeded onto NGM plates. Plates were allowed to dry before the addition of worms (20–30 pretreated adult worms per kill assay plate). The plates were then incubated at 25 °C under aerobic conditions for ETEC and *Salmonella* and under anaerobic conditions for *C. perfringens*. Plates were observed every 24 h until all animals were dead. The number of dead worms out of the total added was recorded. A worm was considered dead when it failed to respond to gentle prodding with a platinum wire. Worms that crawled off the plate or died from bagging were omitted from the analysis.

Experiments were performed three times with three replicates per experiment.

### 2.8. Statistical Analysis

Data from all experiments were analyzed using the Kruskal–Wallis H test (one-way ANOVA by ranks) to determine significant differences between treatment groups. Differences were considered statistically significant at *p* < 0.05. All statistical analyses were conducted in Graph Pad Prism Software (version 9).

## 3. Results

### 3.1. Cell-Free Supernatant from B. velezensis LSSA01, 15AP4 and 2084 Inhibited the Growth of ETEC, C. perfringens and Salmonella Isolates

The percentage inhibition of selected isolates of enterotoxigenic *E. coli*, *C. perfringens* and *Salmonella* spp. by CFS from *B. velezensis* strains after 16 h incubation (LSSA01, 15AP4, 2084 are strains from Enviva^®^ Pro, and 27 is another *B. velezensis* strain introduced for comparison, see details in M&M) is shown in Figure 1A–G. Cell-free supernatant from *B. velezensis* strains LSSA01, 15AP4 and 2084 inhibited the growth of both F4 and F18 ETEC isolates (growth inhibition 36.9–53.2%; Figure 1A,B, respectively). This inhibition was significantly greater than that achieved by CFS from *B. velezensis* strain 27 (*p* < 0.05), which had only a marginal inhibitory effect on growth of the ETEC isolates (growth inhibition 6.3% and 0.8%, respectively; Figure 1A,B). In addition, CFS from *B. velezensis* strains LSSA01, 15AP4 and 2084 inhibited *C. perfringens* type A (growth inhibition 77.5%, 50.2% and 23.6%, respectively), whilst CFS from strain 27 had almost no effect (4.0%, Figure 1C); *C. perfringens* type C was inhibited by strains LSSA01 and 15AP4 (60.8% and 32.9%), whereas strains 2084 and 27 had little effect (growth inhibition −2.3% and −2.6%, respectively, Figure 1D). Finally, the growth of *S. typhimurium*, *S. livingstone* and an unidentified *Salmonella* spp. isolate was inhibited by strains LSSA01, 15AP4 and 2084, but not by strain 27 (growth inhibition 47.8–54.2% for LSSA01, 47.8–51.9% for 15AP4 and 45.9–52.5% for 2084, across *Salmonella* spp. isolates; Figure 1E–G). In all cases, inhibition of *Salmonella* spp. by CFS from LSSA01, 15AP4 and 2084 was significantly greater than that achieved by CFS from strain 27 (*p* < 0.05).

### 3.2. Upregulation of IPEC-J2 Expression of Cytokines and Tight Junction Proteins by B. velezensis Cell-Free Supernatant

The effect of CFS from *B. velezensis* strains LSSA01, 15AP4 and 2084 compared to the medium control (TSB without *B. velezensis* CFS) on the expression of cytokines and tight junction proteins by IPEC-J2 cells is shown in Figure 2A–I. There was no effect of CFS from any *B. velezensis* strain on the expression of IL-6 (Figure 2A), whereas CFS from strain 2084 increased IL-8 production (+12.0% vs. the medium control; *p* < 0.05; Figure 2B). The expression of ZO-1 and ZO-2 was significantly increased by CFS from *B. velezensis* strains LSSA01 and 2084 (+43.1% and +37.9% for ZO-1 and +11.9% and +11.7% for ZO-2, respectively, vs. the medium control; *p* < 0.05; Figure 2C,D), whereas the expression of ZO-3 was enhanced by CFS from all three strains (+31.3%, +17.5% and +28.6% for LSSA01, 15AP3 and 2084, respectively, vs. the medium control; *p* < 0.05; Figure 2E). Furthermore, CFS from *B. velezensis* strain LSSA01 increased (*p* < 0.05) the expression of all claudins and occludins, strain 2084 increased (*p* < 0.05) the expression of claudin-4 and occludin and strain 15AP4 increased (*p* < 0.05) the expression of claudin-4, vs. the medium control (Figure 2F–I).

### 3.3. Cell-Free Supernatant from B. velezensis Strain LSSA01 Facilitated Wound Healing

The effect of CFS from *B. velezensis* strain LSSA01 on simulated wound healing (expressed as the percentage of the wounded area that was visibly recovered) of IPEC-J2 cells during a 96 h experiment window is shown in Figure 3. After 48 h of co-culture with CFS from *B. velezensis* strain LSSA01, IPEC-J2 cells exhibited a significantly larger recovery area than the medium control (recovery area: 43.1 ± 4.3% in LSSA01 vs. 32.3 ± 6.3% in medium control; *p* < 0.05). The thymol and cinnamaldehyde blend (positive control) also significantly improved wound recovery compared to the medium control (*p* < 0.05). There was no effect of CFS from *B. velezensis* strain 15AP4 or 2084 on wound recovery relative to the medium control.

### 3.4. B. velezensis Colony-Forming Units Excluded ETEC Adhesion to IPEC-J2 Cells

The binding affinity of *B. velezensis* strains and ETEC isolates to IPEC-J2 cells after 30 min of co-culture is shown in Table 1. The binding affinity of all strains was relatively low (<5%). However, the binding affinity of the three *B. velezensis* strains was higher than that of all ETEC strains (range 0.216% to 2.98% vs. range 0.000% to 0.172%). There was no apparent link between the genotype and binding affinity of ETEC isolates.

The exclusion assay and competitive exclusion assay results are shown in Figure 4A–D. When IPEC-J2 cells were pretreated with *B. velezensis* CFU, there was a reduction in the adherence of ETEC CFU to the epithelial cells compared to the control (IPEC-J2 cells exposed to ETEC without *B. velezensis*), expressed as the percentage of ETEC cells excluded relative to the total number of ETEC cells loaded onto the cells. Pretreatment of IPEC-J2 cells with *B. velezensis* strain LSSA01 excluded 79.2 ± 6.2% of EC-88, 93.6 ± 0.2% of EC-90 and 91.3 ± 1.5% of EC-23 from adhering to IPEC-J2 cells, respectively (Figure 4A). *B. velezensis* strain 15AP4 had a similar pattern of effect, whereas strain 2084 reduced the adherence of EC-90 and EC-23 but had no effect on EC-88 (*p* < 0.05; Figure 4A). Compared to the control, when IPEC-J2 cells were treated simultaneously with *B. velezensis* and ETEC, there was a significant reduction in ETEC adhesion (Figure 4B). *B. velezensis* strain LSSA01 excluded 98.8 ± 0.2% of EC-88 cells, 93.4 ± 5.7% of EC-23 cells and 95.1 ± 1.1% of EC-90 cells from adhering to IPEC-J2 cells, respectively, and the reductions relative to the control were statistically significant in all three cases (*p* < 0.05; Figure 4B). A similar pattern of effect was observed for *B. velezensis* strains 15AP4 and 2084 (Figure 4B). The effect of *B. velezensis* strains LSSA01 and 15AP4 on exclusion and competitive exclusion of ETEC isolates from binding to IPEC-J2 cells was further investigated using three additional farm-derived isolates of ETEC (with genotypes F4^+^F18^−^, F4^−^F18^+^ and F4^−^F18^−^, respectively; Figure 4C,D). *B. velezensis* strain LSSA01 exhibited a strong exclusion capability against these ETEC field isolates (>75.0% of ETEC cells excluded by LSSA01 and 28.1–95.0% excluded by 15AP4; Figure 4C). In the competitive exclusion assay using these field ETEC isolates, *B. velezensis* strains LSSA01 and 15AP4 each excluded >50% of ETEC cells from binding with IPEC-J2 cells (Figure 4D).

### 3.5. Effect of B. velezensis Strains on C. elegans Survival When Exposed to Pathogenic Bacteria

The mean survival time of worms pretreated with *B. velezensis* strain LSSA01, 15AP4 or 2084 and then exposed to ETEC (F4, F18 or F4^+^F18^−^), *C. perfringens* (type A or C) or *Salmonella* spp. (*S. typhimurium* or *S. livingstone*) over 30 d is shown in Figure 5. Supporting data and statistical analyses are presented in Appendix A along with results of testing of a wider range of pathogen isolates. Worms that had not been pretreated with *B. velezensis* were observed as a control (OP50). The mean survival time varied between *B. velezensis* strains and across pathogen isolates, as shown by the separation of the lines on the graphs in Figure 5A–G. The survival time was significantly longer (*p* < 0.05) in worms pretreated with *B. velezensis* strain LSSA01, 15AP4 or 2084 than in control worms when subsequently exposed to any of the tested pathogens (LSSA01, 15AP4 and 2084 lines on all graphs consistently to the right of lines for control-fed worms; Figure 5A–G and Appendix A). In the presence of ETEC (strain F4), *C. elegans* pretreated with either 15AP4 or 2084 had mean survival times of 15.02 ± 0.80 d and 14.83 ± 0.50 d, respectively, surviving (on average) 35% longer than OP50-fed control animals (Figure 5A, for details see Appendix A). Similar results were observed from pretreatment with the *B. velezensis* strains followed by exposure to *C. perfringens* or *Salmonella* spp. (Figure 5D–G); worms pretreated with *B. velezensis* strains and subsequently exposed to *C. perfringens* or *Salmonella* spp. isolates had a mean survival time of approximately 15–19 d and outlived OP50-fed control worms by an average of 50–60% (Appendix A).

## 4. Discussion

It has previously been shown that feeding *Bacillus*-based probiotics can reduce the prevalence of *E. coli*, *Salmonella*, *Campylobacter* and *C. perfringens* in the swine gut [28], suggesting an antimicrobial effect of *Bacillus*. Other authors [29,30] have also reported antimicrobial effects of *Bacillus*-based probiotics. Data generated from the present study are consistent with this and indicate an antimicrobial effect of CFS from the tested *B. velezensis* strains (LSSA01, 15AP4 and 2084) against common swine pathogens. The general growth inhibitory effect of CFS from these strains was evident against all tested isolates of ETEC (F4 or F18), as well as *C. perfringens* (types A and C) and *Salmonella* spp. (*S. typhimurium*, *S. livingstone* and an unidentified *Salmonella* spp. isolate). The exact antimicrobial compounds within the CFS responsible for the inhibitory effect are unknown and further studies are warranted to identify them. In the wider probiotic literature, studies of *Lactobacillus* spp. and *Pediococcus* spp. have implicated organic acids and bacteriocins acting in synergy as being responsible for their antimicrobial effects in the swine context [31,32,33]. Xie et al. [34] identified an antimicrobial peptide ‘porcine β-defensin 129’ from porcine gut epithelial cells, which had an inhibitory effect on bacterial endotoxin. However, in vivo, the situation is more complicated. It is uncertain whether *Bacillus* could produce bacteriocin and antimicrobial peptides (AMPs) in sufficient quantities in vivo, as their yields are highly dependent on the microflora environment (as a result of quorum sensing among resident bacteria in the microbiota [35]). Further ex vivo quantification of antimicrobial peptides in digesta and feces in *B. velezensis*-supplemented piglets is needed to validate these in vitro results.

Maintaining immune homeostasis is critical for a healthy gut and requires a balanced cytokine profile in which there is equilibrium between anti-inflammatory and pro-inflammatory cytokines. Post-weaning diarrhea is characterized by excessive intestinal inflammation and elevated production of inflammatory cytokines such as IL-1, IL-6 and TNF. Studies of changes in cytokine expression during and after the process of weaning when there is a natural inflammatory response suggest that the resolution of PWD is accompanied by downregulation of pro-inflammatory cytokine production [36]. In the present study, CFS from the tested *B. velezensis* strains had a limited effect on cytokine expression by IPEC-J2 cells, although expression of IL-8 was increased by CFS from strain 2084 (relative to the control). As only three cytokines were measured, the results should not be taken as attributable to all pro-inflammatory cytokines. Interleukin-8 (IL-8) is a leukocyte chemoattractant and stimulant, an alarmin that initiates the innate immune response [37]. Thus, it is hypothesized that the upregulation of IL-8 by CFS from *B. velezensis* strain 2084 may have stimulated the innate immune response, which could act as a protective response in the presence of pathogen infection. However, excessive IL-8 can decrease epithelial barrier integrity by downregulating adhesion junction and tight junction proteins such as OCLN and ZO-1 [38]. There was no evidence of this potential detrimental effect in the present study as no changes in ZO, claudin or OCLN expression were observed in IPEC-J2 cells following pretreatment with *B. velezensis* CFS.

In PWD, there is a disturbance in the integrity and permeability of the intestinal cell layer. Barrier integrity is maintained by intracellular adhesion complexes including transmembrane tight junction (TJ) proteins [39]. Dysregulation of TJ proteins such as ZO, claudins and OCN can lead to increased permeability, predisposing piglets to ETEC infection by allowing ETEC fimbriae F4 and F18 to adhere to the intestinal epithelial cell layer. Enhancing the intestinal immune defense system is crucial to ensuring optimal health and the prevention of uncontrolled inflammation in piglets [40]. Zona occludens proteins are known to interact with claudins and are required for TJ assembly. Cells that fail to express ZO-1 and ZO-2 are not able to properly assemble claudins to the TJ and exhibit reduced barrier function [41,42]. Similarly, decreased expression levels of claudins in the intestinal epithelium have been associated with reduced barrier integrity [43]. Gut barrier disruption (commonly known as a “leaky gut”) facilitates the entry of pathogenic bacteria and bacterial toxins into the systemic circulation, where they provoke systemic inflammation and trigger diseases such as PWD [44]. The present study results showed that treatment with CFS from one or more of the tested *B. velezensis* strains (LSSA01, 15AP4 and 2084) increased the expression of ZO, claudins and OCLN in IPEC-J2 cells. These findings lead us to predict that CFS from the probiotic *B. velezensis* strains may have the capacity to improve gut barrier integrity, as has previously been shown for several other probiotic strains [45,46]. It may further be predicted that any beneficial effect of *B. velezensis* CFS on barrier integrity would likely be maximized if the probiotic was administered to piglets prophylactically, before disease onset.

Our results also indicated a beneficial effect of CFS from one of the tested *B. velezensis* strains (LSSA01) on IPEC-J2 cell wound healing. In animals, upon injury of the intestinal epithelial cell layer, there is a cascade of response orchestrated by the epithelium itself as well as by temporal recruitment of immune cells into the wound bed, which together function to initiate the wound-healing process [47]. Our results have shown that CFS from strain LSSA01-assisted IPEC-J2 epithelial cells achieve more rapid reparation following a simulated ‘wound’ than control, untreated, cells, suggesting that intact IPEC-J2 cells may have migrated along the exposed basement membrane to fill in the area where cells were damaged and restore epithelial barrier integrity.

Pathogen inhibition, cell cytokines and wound-healing assays are all CFS-based. The question remaining unanswered is how live bacteria perform in these assays. This is restrained by the limitation of the setup of the cell model. Indeed, live bacteria and its CFS may represent two different mechanisms that confer effects in the gut. On the one hand, the specific metabolic profiles of probiotic-derived CFS encompass a variety of organic acids, bacteriocins and other bioactive compounds, which collectively contribute to their activities against pathogenic bacteria and to improved immune responses [48]. On the other hand, there is no doubt that intestinal epithelial cells recognize luminal bacterial signals by a variety of pattern recognition receptors including TLRs and nod-like receptors (NLRs). The signaling leads to the expression of inflammatory cytokines and chemokines such as IL-1β, IL-6, IL-18 and CCL20. The cells and cytokines produced establish part of the innate immune system, serving as the first-line defense system in a host [49]. By utilizing intestinal organoid, organ-on-a-chip or animal models, further efforts could be made to investigate the direct host–microbe interaction.

Rapid intestinal colonization of ETEC is critical in the pathogenesis of PWD [50]. Adhesion of bacterial cells to the epithelium is a crucial step in colonization. Our results showed that the three tested *B. velezensis* strains had a higher binding affinity to IPEC-J2 cells than the tested ETEC isolates. Further, we observed that pretreatment of IPEC-J2 cells with CFU from *B. velezensis* reduced the adhesion of ETEC CFU to IPEC-J2 cells (increased the ETEC exclusion percentage). The major ETEC fimbriae cell-binding receptors are the F4 receptors MUC4 and APN [51] and the F18R receptor [52], which has been shown to be synthesized by A1-fucosyltransferase (FUT1) and FUT2 [53]. However, in the present study, the ETEC exclusion effect appeared to be independent of fimbriae receptor expression because it was equally observed with exposure to F4^−^F18^−^ ETEC cells as with exposure to F4^+^F18^−^ and F4^−^F18^+^. As preliminary experimental work in our laboratory showed that F4 and F18 receptor expression by IPEC-J2 cells was unaltered by co-culture with *B. velezensis* CFU (see Appendix A), these results raise the question as to whether ETEC may have the capacity to invade epithelial cells by mechanism(s) other than through attachment via the fimbriae receptors. In fact, in humans, a significant number of ETEC strains that do not express identifiable fimbriae are known to cause infection, and two chromosomally borne invasion loci, designated tia and tib (toxigenic invasion loci A and B), have been identified and implicated in this effect [54,55]. It has been proposed that tia and tib encode proteins that mediate the adherence and invasion of *E. coli* strains containing recombinant Tia or TibA proteins to intestinal epithelial cells [53,54,55,56]. Further studies are warranted to determine how *B. velezensis* excluded ETEC isolates from adhering to IPEC-J2 cells and whether this also happens in vivo.

Pretreatment of adult L4-stage *C. elegans* worms with CFU of *B. velezensis* strain LSSA01, 15AP4 or 2084 extended their average lifespan significantly, with a greater beneficial effect seen with LSSA01 than 15AP4 or 2084. Similar improvements to *C. elegans* longevity have been reported under stress conditions with worms fed *B. subtilis* cells [55] and under challenge conditions with *S. enterica* or *Staphylococcus aureus* in worms fed lactic acid-producing bacteria [57]. The mechanism of these effects is unclear. However, it has been shown that the innate immune system of *C. elegans* is involved in its response to probiotic bacteria [58,59] and several immune pathways have been implicated: the TGF-β pathway [60], the p38 MAPK pathway [61] and the insulin-like signaling (ILS) pathway [62]. The existing literature together with our observations that *B. velezensis* CFS increases TJ protein expression by IPEC-J2 cells suggest that an improvement in the host innate immune response may be one mechanism via which *B. velezensis* CFS increased *C. elegans* survival. This requires further testing in vitro and in vivo to determine whether these effects are also observed in piglets.

## 5. Conclusions

This is the first in vitro study to evaluate the effect of probiotic *B. velezensis* on host responses to pathogenic bacteria that lead to PWD in swine. Our findings have shown that CFS from all of the tested *B. velezensis* strains (LSSA01, 15AP4 and 2084) exhibited antimicrobial properties against pathogenic F4 and F18 ETEC isolates and that CFS from one or more of the strains was also effective at inhibiting the growth of *C. perfringens* and *Salmonella* spp. isolates. In addition, CFS from one or more of the *B. velezensis* strains exhibited immunomodulatory capabilities (increased expression of IL-8, claudins and occludin) when administered to IPEC-J2 cells. Live colony forming units of all three probiotic strains exhibited a stronger binding capacity to IPEC-J2 cells than the tested ETEC isolates and excluded the pathogen from binding to these cells when incubated in co-culture (to varying extents). Further, CFS from one strain, LSSA01, improved simulated ‘wound healing’ of an IPEC-J2 monolayer in comparison to an untreated control, in a 96 h in vitro assay. Finally, pretreatment of live, adult, *C. elegans* worms with CFU from any of the three *B. velezensis* strains extended *C. elegans* survival when subsequently exposed to ETEC, *C. perfringens* or *Salmonella* spp. isolates over a 30 d experimental period. These results highlight the potential for the use of the probiotic strains in swine for the control of pathogens that cause PWD. It is of note that current cell and *C. elegans* studies are limited as they cannot fully model how probiotics may interact with all the molecules and cell types that exist within a complex organ or host. Probiotics can only be tested in isolation or a simplified model, whereas the host is a dynamic environment where numerous pathways and cells are in constant communication. This can make it difficult for in vitro studies to predict the complexities of potential interactions. Therefore, further in vitro and in vivo testing is warranted to explore and confirm the effects in a farm setting. Future studies should consider whether there is synergy among the three strains in their beneficial effects and whether their effects extend to other swine pathogens than those considered here.

## Figures and Tables

**Figure 1 microorganisms-13-01247-f001:**
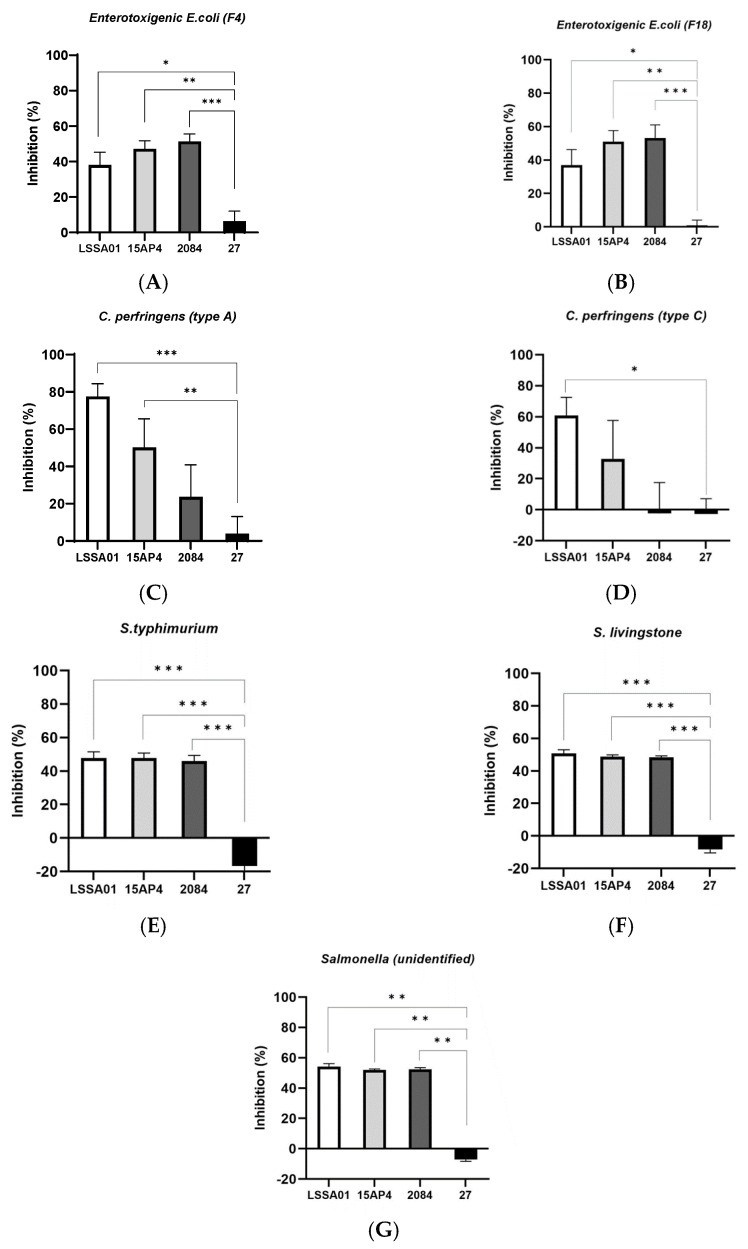
Effect of cell-free supernatant (CFS) from *B. velezensis* strains LSSA01, 15AP4, 2084 and 27 on the growth of ETEC isolates (**A**) F4 and (**B**) F18, *C. perfringens* isolates (**C**) Type A and (**D**) Type C, *Salmonella typhimurium* (**E**), *S. livingstone* (**F**) and an unidentified *Salmonella* spp. isolate (**G**) after 16 h of incubation. Values are shown as means ± standard deviation (SD). Experiments were performed three times with three replicates per experiment. Details of the pathogens included in the assay are listed in Appendix A. *: *p* < 0.05; **: *p* < 0.01; ***: *p* < 0.001.

**Figure 2 microorganisms-13-01247-f002:**
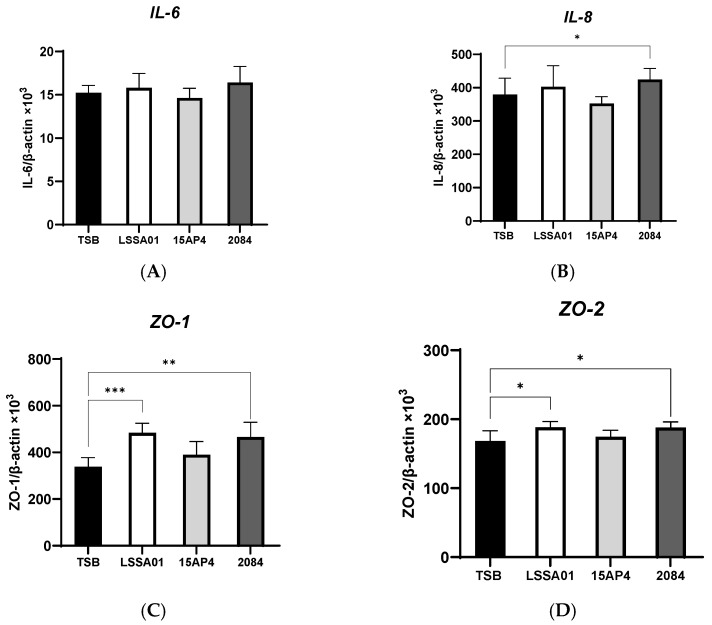
Effect of CFS from *B. velezensis* strains LSSA01, 15AP4 and 2084 compared to a control (culture medium only) on the IPEC-J2 cell expression of (**A**) IL-6, (**B**) IL-8, (**C**) ZO-1, (**D**) ZO-2, (**E**) ZO-3, (**F**), claudin-1, (**G**) claudin-3, (**H**) claudin-4 and (**I**) occludin, after 6 h incubation. Data were normalized to two sets of housekeeping genes using the following equation: Value = 2 − (Ct sample − Ct housekeeping) × 10^3^. Experiments were performed two times with four replicates per experiment (eight replicates in total). Values represent means and associated SD. *: *p* < 0.05; **: *p* < 0.01; ***: *p* < 0.001.

**Figure 3 microorganisms-13-01247-f003:**
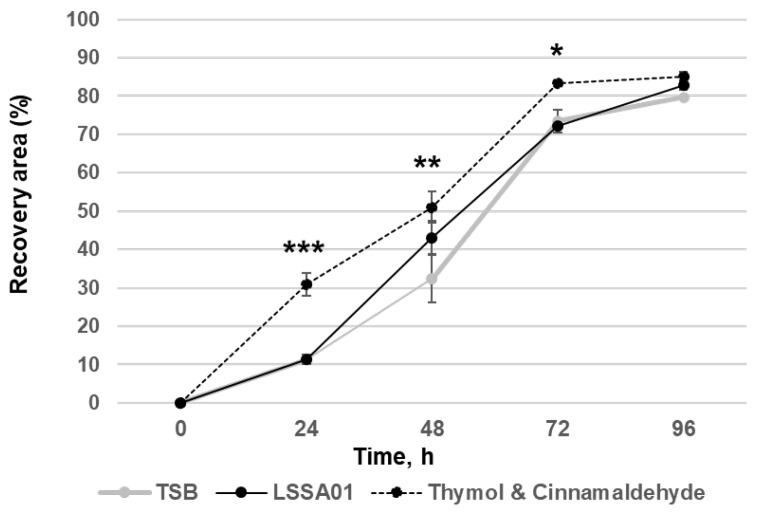
Effect of CFS from *B. velezensis* strain LSSA01 compared to a TSB medium control (negative control) and a thymol and cinnamaldehyde blend (positive control) on IPEC-J2 cell wound healing over 96 h. The data are mean values with associated standard error (SE) bars. The data represent two experiments with three replicates per experiment. *: *p* < 0.05; **: *p* < 0.01; ***: *p* < 0.001.

**Figure 4 microorganisms-13-01247-f004:**
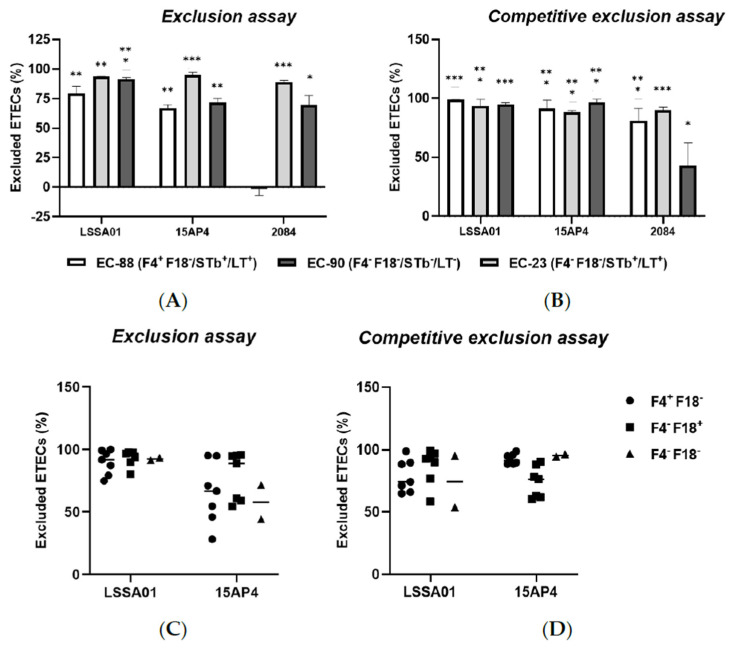
The percentage exclusion of ETEC isolates (EC-88, EC90 and EC-23) from adhering to IPEC-J2 cells following (**A**) pretreatment or (**B**) simultaneous treatment of IPEC-J2 cells with *B. velezensis* strain LSSA01, 15AP4 or 2084. In addition, the percentage exclusion of three ETEC field-derived isolates (of genotypes F4^+^F18^−^, F4^−^F18^+^ and F4^−^F18^−^, respectively) by (**C**) pretreatment or (**D**) simultaneous treatment of IPEC-J2 cells with CFU of *B. velezensis* strain LSSA01 or 15AP4 is shown. In (**A**,**B**), the data are mean values with associated SE and are drawn from two experiments with seven replicates in total; In (**C**,**D**), the horizontal bar represents the median value of the data set. *: *p* < 0.05; **: *p* < 0.01; ***: *p* < 0.001. Statistical comparisons are compared to the response in the control (IPEC-J2 cells exposed to ETEC without *B. velezensis*).

**Figure 5 microorganisms-13-01247-f005:**
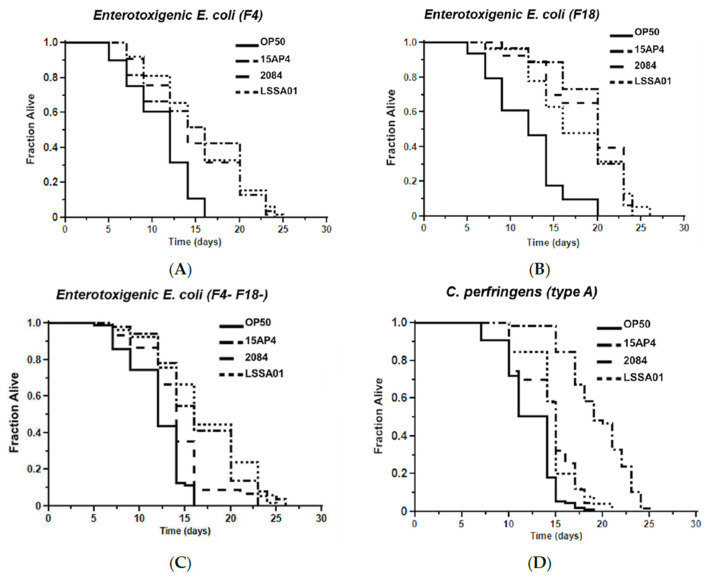
Effect of 48 h pretreatment of *C. elegans* worms with CFU of *B. velezensis* strains LSSA01, 15AP4 or 2084 on survival time over 30 d when subsequently exposed to pathogenic isolates of ETEC (**A**) F4, (**B**) F18, (**C**) F4^+^F18^−^ (**D**) *C. perfringens* type A, (**E**) *C. perfringens* type B, (**F**) *S. typhimurium* and (**G**) *S. livingstone*. The data represent the mean of three experiments each with three replicates. (Results from a wider sample of ETEC, *C. perfringens* and *Salmonella* spp. isolates and accompanying statistical analyses are presented in Appendix A.)

**Table 1 microorganisms-13-01247-t001:** Binding affinity of *B. velezensis* strains and ETEC isolates to IPEC-J2 cells after 30 min of co-culture. The data are mean values drawn from two experiments with three replicates per experiment.

	Ecotoxigenic *Escherichia coli* (ETEC) Isolate F4 and F18 Fimbriae Genotype	Binding Affinity (%)
*B. velezensis* strain		
LSSA01	Not applicable	0.298
15AP4	Not applicable	0.216
2084	Not applicable	2.607
*ETEC* isolate		
EC-23	F4^−^F18^+^	0.003
EC-50	F4^−^F18^+^	0.009
EC-58	F4^−^F18^+^	0.053
EC-61	F4^−^F18^+^	0.052
EC-62	F4^−^F18^+^	0.009
EC-63	F4^−^F18^+^	0.000
EC-65	F4^−^F18^−^	0.006
EC-88	F4^+^F18^−^	0.172
EC-89	F4^+^F18^−^	0.000
EC-90	F4^+^F18^−^	0.001
EC-91	F4^−^F18^−^	0.002
EC-92	F4^+^F18^−^	0.145

## Data Availability

The original contributions presented in this study are included in the article. Further inquiries can be directed to the corresponding author.

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
