# Peer review of "Evaluation of Probiotic Bacillus velezensis for the Control of Pathogens That Cause Post-Weaning Diarrhea in Piglets—Results from In Vitro Testing and an In Vivo Model Using Caenorhabditis elegans"

_microorganisms, 2025, doi:10.3390/microorganisms13061247_

Round 1
Reviewer 1 Report
Comments and Suggestions for Authors
General Comments:
The manuscript investigates the antimicrobial potential of Bacillus velezensis strains against swine pathogens. The authors used in vitro models to assess probiotic candidates in relation to intestinal integrity and employed a nematode model to evaluate protective effects against pathogen challenge. While the study is well-structured and addresses a relevant topic, a limitation statement is necessary to acknowledge the uncertainty regarding how these findings may translate to in vivo conditions.
Specific Comments:
-
Line 104: Please clarify the origin of the probiotic strain. From what source was it originally isolated?
-
Line 125: Specify the growth conditions used for each bacterial strain.
-
Line 132: What oxygen conditions were considered appropriate for each pathogen during incubation?
-
Line 134: Why were cell-free supernatants (CFS) used instead of live bacteria? Please justify this choice.
-
Additional: Was the pH of the CFS measured? If so, please report the values.
-
Line 181: The author’s name appears to be missing in the citation—please correct this.
-
Line 288: It would be helpful to briefly reintroduce or redefine strain 27 at this point in the manuscript for clarity.
-
Figure/Table (F4 E. coli inhibition results): Please re-examine the results. The CFS from strain 15AP4 shows lower inhibition against F4 E. coli compared to strain 2084, yet has more significance indicators (asterisks). This inconsistency needs clarification.
-
Line 391: Table 1 suggests that ETEC binding affinity to IPEC-J2 cells is low, and B. velezensis strains also exhibit low binding affinity. Could this be due to insufficient incubation time, preventing bacterial colonization? If so, how might this affect the conclusions drawn from the competitive exclusion assay?
-
Line 520: Could the lack of observed effect from B. velezensis be attributed to the use of CFS rather than live bacteria? Please discuss this possibility.
-
Line 573: Do IPEC-J2 cells express F4 and F18 receptors? Clarifying this would help interpret results related to ETEC adhesion.
-
Line 589: Please remove the redundant word “that.”
Author Response
Reply to Reviewer 1:
General Comments:
The manuscript investigates the antimicrobial potential of Bacillus velezensis strains against swine pathogens. The authors used in vitro models to assess probiotic candidates in relation to intestinal integrity and employed a nematode model to evaluate protective effects against pathogen challenge. While the study is well-structured and addresses a relevant topic, a limitation statement is necessary to acknowledge the uncertainty regarding how these findings may translate to in vivo conditions.
We are greatly appreciated of review’s effort and fully agree with the reviewer’s comment. To the limitation of the in vitro cell models, we have added a text related to in vitro and in vivo translation in the Discussion (line 652-657)
Related to in vivo translation of this product, we have just finished 2 successful trials, as evidenced by significant lower diarrhea frequency or mortality, significantly improved average daily gain ADG and body weight. We are under preparation of another separate paper now. Abundant beneficial gut bacteria in treated piglets encourages us to further explore its underlying mechanisms.
Specific Comments:
- Line 104:Please clarify the origin of the probiotic strain. From what source was it originally isolated? Thanks for reviewer’s reminder. LSSA01 and 15AP4 were isolated from turkey litter by Agtech Products (KS, USA) in 2002 and 2000 respectively. 2084 is a non-proprietary commercial strain from Novonesis (Bagsværd, Denmark) and Microbial Discovery Group (WI, USA) in 2015. We have added this information (line 79-81, line 103-110, ref 22-25 in the text). The genome sequencing and assay validation has been patented in PCT/US2O13/02683O; The safety and efficacy of the strains using in chicken and turkey has been documented at EFSA in 2016 and 2022 respectively; Its efficacy and application in swine has been patented (INV230049USPSP2) in 2024.
- Line 125:Specify the growth conditions used for each bacterial strain. Thanks for the comment. Wording has been changed to improve clarity (line 136).
- Line 132:What oxygen conditions were considered appropriate for each pathogen during incubation? Thanks for the comment. The bacteria growth condition was incubated consistently all along the assay: ETECs and Salmonella in anaerobic condition; CP in anaerobic condition (line140-143). We have removed the “oxygen” expression to avoid confusion (line 149-150).
- Line 134:Why were cell-free supernatants (CFS) used instead of live bacteria? Please justify this choice.
Thanks for the comment. We have added a text in the discussion to clarify rationale of current study as well as further study direction (line 585-598).
Overall, we believe that live bacteria and its CFS represent 2 perspectives to confer their effects in gut. On the one hand, the specific metabolic profiles of probiotic-derived cell free supernatant (CFS) encompass a variety of organic acids, bacteriocins, and other bioactive compounds that collectively contribute to their antimicrobial activity against pathogenic bacteria (Barros CP, 2020), and to improve immune responses. The priority of this study is to validate the product’s inhibitory effect and then further identify the bioactive elements. The latter part in still ongoing.
On the other hand, there is no doubts that intestinal epithelial cells recognize luminal bacterial signals by a variety of pattern recognition receptors including TLRs and nod-like receptors (NLRs). The signalling leads to the expression of inflammatory cytokines and chemokines such as IL1β, IL-6, IL-18, and CCL20. The cells and cytokines produced are established as part of the innate immune system serving as the first line defence system in host (Li D, 2021). In a preliminary experiment, we have cocultured probiotics live bacteria with IPEC- J2 cells for 30 mins till 2 hours. Cell lysates have been tested for PCR for the expression of F4/ F18 receptor, pIgR, tight junction (occluding and claudin) transcriptions factors (NFkB) and cytokines (IL-6, IL-8, G-CSF and TGF-β), Compared to medium control, there is no evident differences (<10%). It is unclear whether this is due to cell culture condition, or incubation time. Further effort and other experimental setup (which may better resemble gut microenvironment, i.e intestinal organoid or animal study is needed to further investigate bacteria colonisation, proliferation, host-microbe interaction and the disease pathogenesis.
Additional: Was the pH of the CFS measured? If so, please report the values. Thanks for the question. pH was not measured in this study. In a previous study, we have measured them and they were between 7.65-7.85 among all CFS samples. As strain 27, the same B. velezensis was introduced in the assay and has been found almost no inhibitory effect, it is unlikely that low pH (derived from acid production) plays a role in the inhibitory effect. Metabolomic profiling is currently ongoing.
Line 181: The author’s name appears to be missing in the citation—please correct this. Thanks. The missing author’s name has been added (line 199).
- Line 288:It would be helpful to briefly reintroduce or redefine strain 27 at this point in the manuscript for clarity. Thanks for the comment. The brief introduction of 27 has been added in text (line 308-309).
- Figure/Table (F4 E. coli inhibition results):Please re-examine the results. The CFS from strain 15AP4 shows lower inhibition against F4 E. coli compared to strain 2084, yet has more significance indicators (asterisks). This inconsistency needs clarification. Thanks for pointing out this mistake. We have cross-checked the original data and found it due to wrong labelling between 15AP4 and 2084. We have corrected the mistake.
- Line 391:Table 1 suggests that ETEC binding affinity to IPEC-J2 cells is low, and velezensis strains also exhibit low binding affinity. Could this be due to insufficient incubation time, preventing bacterial colonization? If so, how might this affect the conclusions drawn from the competitive exclusion assay? Thanks for this great comment. The effect of the incubation time on the adhesion of probiotics has not been fully investigated. In most assays, probiotics are incubated for half hour. Some investigators have reported little or no effect of incubation time on the level of adhesion (Aissi EA, 2001; Ouwehand AC, 2002), while some others stated that incubation time may have a major influence on the observed adhesion (Ouwehand AC, 2003). This discrepancy might due to different bacteria speciecs, but can also be explained by sedimentation of the microbes in assay procedure: as bacteria-cell incubation progresses, more and more bacteria sediment and leads to artificially increased levels of adhesion. Since sedimentation is not likely to happen in vivo, we believe an exposure of 15-30 min in the upper small intestine is physiologically relevant contact time between bacteria and the intestinal mucosa. Thus, we applied bacteria to cell and cultured them to adhere for 30 minutes.
Line 520: Could the lack of observed effect from B. velezensis be attributed to the use of CFS rather than live bacteria? Please discuss this possibility. Thanks for the comment. We have added a text in the discussion to clarify rationale of current study as well as further study direction (line 585-598).
Overall, we believe that live bacteria and its CFS represent 2 perspectives to confer their effects in gut. On the one hand, the specific metabolic profiles of probiotic-derived cell free supernatant (CFS) encompass a variety of organic acids, bacteriocins, and other bioactive compounds that collectively contribute to their antimicrobial activity against pathogenic bacteria (Barros CP, 2020). The scope and the priority of this study is to validate the product’s inhibitory effect and then further identify the bioactive elements. The latter part in still ongoing.
On the other hand, there is no doubts that intestinal epithelial cells recognize luminal bacterial signals by a variety of pattern recognition receptors including TLRs and nod-like receptors (NLRs). The signalling leads to the expression of inflammatory cytokines and chemokines such as IL1β, IL-6, IL-18, and CCL20. The cells and cytokines produced are established as part of the innate immune system serving as the first line defence system in host (Li D, 2021). In a preliminary experiment, we have cocultured probiotics live bacteria with IPEC- J2 cells for 30 mins till 2 hours. Cell lysates have been tested for PCR for the expression of F4/ F18 receptor, pIgR, tight junction (occluding and claudin) transcriptions factors (NFkB) and cytokines (IL-6, IL-8, G-CSF and TGF-β), Compared to medium control, there is no evident differences (<10%). It is unclear whether this is due to cell culture condition, or incubation time ( 30 minutes). Further effort and other experimental set up (which may better resemble gut microenvironment, i.e intestinal organoid or animal study) is needed to further investigate bacteria colonisation, proliferation, host-microbe interaction and the disease pathogenesis.
Line 573: Do IPEC-J2 cells express F4 and F18 receptors? Clarifying this would help interpret results related to ETEC adhesion. Yes, they do. We have examined it in a preliminary experiment and found that the cells do express F4 and F18 receptors. We have added this info into supplementary data (Figure S1) and adjust the text (line 611) and disclose the data.
- Line 589:Please remove the redundant word “that.” The redundant “that” has been removed (line 627). Thanks to the reviewer.
References
- Chen WC. Juang RS and Wei YH., 2015 Applications of a lipopeptide biosurfactant, surfactin, produced by microorganisms. Biochem. Engin. J. 103, 158-169. 10.1016 / j.bej.2015.07.009.
- Aissi EA, Lecocq M, Brassart C, Bouquelet S. Adhesion of some Bifidobacteria strains to human enterocyte-like cells and binding to mucosal glycoproteins. Microb Ecol Health Dis 2001; 13: 32/9.
- Ouwehand AC, Suomalainen T, Tolkko S, Salminen S. In vitro adhesion of propionic acid bacteria to human intestinal mucus. Lait 2002; 82: 123/30.
- Ouwehand AC and Salminen S. In vitro Adhesion Assays for Probiotics and their in vivo Relevance: A Review 2003; Microbial Ecology in Health and Disease, 15:4, 175-184, DOI: 10.1080/08910600310019886
- Barros CP, Guimarães JT, Esmerino EA, Duarte MCK, Silva MC, Silva R, et al. Paraprobiotics and postbiotics: concepts and potential applications in dairy products. Curr Opin Food Sci. 2020; 32:1–8. doi: 10.1016/j.cofs.2019.12.003
- Li D, Wu M. Pattern recognition receptors in health and diseases. Signal Transduct Target Ther. 2021; 6:291. doi: 10.1038/s41392-021-00687-0.

Reviewer 2 Report
Comments and Suggestions for Authors
This is a novel study concerning the testing of some new strains of probiotics in piglets, enhancing the decrease of pathogenic bacteria and most importantly reducing diarrhea cases. Apart from the absence of molecular techniques which is a limitation that should be mentioned, the study is complete and comprehensive. The pathways presented are interesting and supported. I congatulate the authors for the nice work done, but I have some minor revisions and a more generic and important comment as follows.
The main comment is the absence of data regarding reduction of diarrhea cases or related data. Such experimentation would be crucial
Minor specific comments
Please check Italics of species names in the whole manuscript
In the introduction there is not any information for the choice of B. velezensis for the study. It is firstly mentioned in the scope paragraph without any previous info. Why were these strains chosen? Also some info regarding the characterization of the three strains is missing.
The SD in the majority of the expressed proteins is very low. Is there any potential commentary on this observation?
Author Response
Reply to Reviewer 2
This is a novel study concerning the testing of some new strains of probiotics in piglets, enhancing the decrease of pathogenic bacteria and most importantly reducing diarrhea cases. Apart from the absence of molecular techniques which is a limitation that should be mentioned, the study is complete and comprehensive. The pathways presented are interesting and supported. I congatulate the authors for the nice work done, but I have some minor revisions and a more generic and important comment as follows.
The main comment is the absence of data regarding reduction of diarrhea cases or related data. Such experimentation would be crucial
To improve translation power of in vitro assays and support in vivo trial is our objective. We acknowledge positive comments from the reviewer and fully agree with the comments. We have added a text related to in vitro and in vivo translation in the Discussion and state the limitation of the current model (line 652-657).
This study starts from in vitro assays to understand disease mechanism, and then validate our hypothesis in piglet trials in parallel. We have just finished 2 successful trials, part of the data has been submitted to Digestive Physiology of Pigs (DPP) 2025 (accepted abstract, #93072), while full papers are under preparation. In brief,
- Trial one (144 weaning piglets, day 1 to 42 post-weaning): Probiotics product reduces diarrhea frequency 6.3% (p<0.05), improve ADG 7.0% (p<0.05), lower FCR4.5% (p<0.05).
- Trial two (400 weaning piglets, day 1 to 42 post-weaning): Overall mortality was reduced with probiotic supplementation (1.6 vs 1.0%, p<0.05) and with similar growth promoting effects.
Along with, improved the abundance of certain beneficial microbial taxa have been observed: such as Segatella on day 21 but not day 0, Clostridium, Faecalibacterium and Xylanibacter on day 42 but not before, were significantly more abundant (p<0.05) in the probiotic-fed diet.
Taken together, the data indicates that introduction of probiotics likely plays a beneficial role and contributes to gut health.
Minor specific comments
Please check Italics of species names in the whole manuscript Thanks for the reminder. We have cross-checked them and corrected them while necessary.
- In the introduction there is not any information for the choice of B. velezensis for the study. It is firstly mentioned in the scope paragraph without any previous info. Why were these strains chosen? Also some info regarding the characterization of the three strains is missing. Thanks for reviewer’s reminder. LSSA01 and 15AP4 were isolated from turkey litter by Agtech Products (KS, USA) in 2002 and 2000 respectively. 2084 is a non-proprietary commercial strain from Novonesis (Bagsværd, Denmark) and Microbial Discovery Group (WI, USA) in 2015. We have added this information (line 79-81, line 103-110, ref 22-25 in the text). The genome sequencing and assay validation has been patented in PCT/US2O13/02683O; The safety and efficacy of the strains using in chicken and turkey has been documented at EFSA in 2016 and 2022 respectively; Its efficacy and application in swine has been patented (INV230049USPSP2) in 2024.
- The SD in the majority of the expressed proteins is very low. Is there any potential commentary on this observation? Thanks for the comments. It is indeed a mistake: instead of SD, all Figure 2 graphs were shown as mean ± SEM. In order to align with other figures, we have changed the graph column to mean ± SD, as originally stated in figure legend.
